# Time point- and plant part-specific changes in phloem exudate metabolites of leaves and ears of wheat in response to drought and effects on aphids

**Jana Stallmann, Caroline A. A. Pons, Rabea Schweiger, Caroline Müller** *

Department of Chemical Ecology, Bielefeld University, Bielefeld, Germany

* caroline.pons@uni-bielefeld.de

**Data Availability Statement:** All relevant data are within the paper and its Supporting information files.

## Abstract

Alterations in the frequency and intensity of drought events are expected due to climate change and might have consequences for plant metabolism and the development of plant antagonists. In this study, the responses of spring wheat (*Triticum aestivum*) and one of its major pests, the aphid *Sitobion avenae*, to different drought regimes were investigated, considering different time points and plant parts. Plants were kept well-watered or subjected to either continuous or pulsed drought. Phloem exudates were collected twice from leaves and once from ears during the growth period and concentrations of amino acids, organic acids and sugars were determined. Population growth and survival of the aphid *S. avenae* were monitored on these plant parts. Relative concentrations of metabolites in the phloem exudates varied with the time point, the plant part as well as the irrigation regime. Pronounced increases in relative concentrations were found for proline, especially in pulsed drought-stressed plants. Moreover, relative concentrations of sucrose were lower in phloem exudates of ears than in those of leaves. The population growth and survival of aphids were decreased on plants subjected to drought and populations grew twice as large on ears compared to leaves. Our study revealed that changes in irrigation frequency and intensity modulate plant-aphid interactions. These effects may at least partly be mediated by changes in the metabolic composition of the phloem sap.

## Introduction

Plants are subjected to various environmental impacts, which influence both their own performance but also the development of organisms interacting with the plants [1–3]. Amongst those impacts, the climatic conditions during growth crucially determine plant development and fitness. Climate extremes such as droughts have become more frequent and intense and are predicted to become even more severe in the coming decades [4]. In agriculture, sufficient frequencies and amounts of precipitation are essential to sustain high crop yields [5, 6]. Along with impacts on plant growth, the metabolic composition of different plant parts can be

**Funding:** This work was funded by a grant of the Deutsche Forschungsgemeinschaft (MU 1829/23-1).

**Competing interests:** The authors have declared that no competing interests exist.

affected by drought [7–9]. For example, plants accumulate certain metabolites to facilitate water uptake and to protect cell structures and functions [10]. Changes in plant physiology and nutritional quality may influence interactions of plants with their biotic environment such as herbivorous insect pests, which in turn can impact crop productivity [11]. Moreover, effects on the metabolome and different drought tolerance traits may depend on the duration of stress [12], the developmental stage of the plant [13] and differ between plant parts [9], but these aspects have rarely been investigated. Drought-induced bottom-up effects on insect herbivores and potential feedbacks of herbivores on plants should also be considered in integrated pest management [14].

Different hypotheses regarding the effects of (drought) stress on the development of insect herbivores have been postulated. The plant vigour hypothesis predicts that insect development is best on plants grown under optimal conditions, explained by a favourable water and nutrient content [15]. In contrast, the plant stress hypothesis states that insects develop best on stressed plants due to a breakdown of proteins and thus an increase of accessible nitrogen in plant tissues [16]. In addition, stressed plants should have fewer resources available to synthesise defences. The pulsed stress hypothesis discriminates between continuous and intermittent stress and states that particularly sap-feeding insects profit from an occasional recovery of cell turgor and a stress-induced increase in accessible nitrogen [17]. There is support for all three hypotheses, depending on the insect species, its feeding guild, developmental stage and the magnitude of stress applied [18–21]. The latter is directly related to the number of drought cycles plants are exposed to [22] but also to the phenology of the plants [23].

The life history of aphids is particularly shaped by the accessibility and quality of the phloem sap of their host plants. Under drought, some plant species show increased concentrations of several amino acids in the phloem sap or phloem exudates [24–26]. Furthermore, sucrose levels can be higher in the phloem sap of plants experiencing drought, while some defence metabolites show lower concentrations [25]. The development of aphids is positively, not or negatively affected by drought [25–27], depending on the plant species as well as the diet breadth of the aphid species. Given this specificity, it may be impossible to generally predict herbivore responses to drought stress of their host plants.

As one of the world's most important crop plant species, wheat (*Triticum aestivum* L., Poaceae) is well-investigated regarding impacts of abiotic environmental factors and pest resistance [28, 29]. Apart from detrimental effects of water scarcity on wheat biomass production, probably more than 20% of the global wheat yield is lost due to pests [29]. One of these pests is the aphid species *Sitobion avenae* F. (Hemiptera: Aphididae), which feeds on phloem sap of wheat [30]. This herbivore proliferates particularly on the inflorescences and can cause severe yield losses by impacting grain filling and by transmitting plant viruses [30]. In a previous publication, we demonstrated that the biomass of wheat plants was lower for drought-exposed plants than for well-watered plants, indicating that drought caused some stress, whereas the applied water use efficiency was higher in drought-stressed plants [31]. Furthermore, the metabolome of wheat flag leaves differed between drought-exposed and well-watered plants, measured at two different time points [31]. Changes in the leaf metabolome were more pronounced in plants exposed to pulsed drought stress compared to continuously drought-stressed plants, although all drought-stressed plants received the same cumulative amount of water [31]. Changes due to these irrigation treatments were also found in the amino acid composition of phloem exudates of wheat leaves, with drastic increases in relative concentrations of proline in plants that have experienced pulsed drought stress [32]. However, it remained unclear how further primary metabolites (organic acids, sugars) in the phloem exudates are affected by drought and whether effects of drought on the phloem exudate composition and on aphids differ between time points as well as between leaves and ears.

The objectives of the present study were to assess effects of different drought regimes on various primary metabolites in phloem exudates of wheat and on aphids, considering different time points and plant parts. We grew well-watered, continuously drought-stressed and pulsed drought-stressed wheat plants, analysed various primary metabolites in phloem exudates of leaves (at day 77 and day 93 after sowing) and ears (at day 93) and conducted bioassays with *S. avenae* on leaves and ears. We hypothesised that the metabolite profiles of phloem exudates are altered by the drought stress treatments, with more prominent changes in plants exposed to pulsed than to continuous drought stress [31, 32]. In particular, we expected compatible solutes such as proline and sugars to increase in relative concentration under drought stress in the phloem exudates of both leaves and ears. For the aphids we predicted a higher population growth and survival on drought-stressed compared to well-watered plants due to higher relative proline concentrations found in leaf phloem exudates of drought-stressed plants [32] and in line with the plant stress hypothesis [16]. We expected that their development is even better on pulsed drought-stressed than on continuously drought-stressed plants, in accordance with the pulsed stress hypothesis [17]. Furthermore, we hypothesised that after ear emergence, aphid populations grow bigger on ears compared to leaves, because nutrients are transported from vegetative to reproductive tissues during grain loading [33].

## Materials and methods

### Plant cultivation

The experiment was carried out in Bielefeld, Germany, from December 2016 to May 2017. Seeds of spring wheat (*Triticum aestivum* cv. Tybalt; von Borries-Eckendorf, Leopoldshöhe, Germany) were germinated in a glasshouse chamber at 22°C, 46% relative humidity (r.h.), light:dark 12:12 h in a 1:1 mixture of soil (Fruhstorfer Pikiererde Type P, Hawita Group, Vechta, Germany) and river sand. The substrate had been steamed at 90°C before use. Six days after sowing, five seedlings were placed in each pot (4 l, 15.7 x 15.7 x 23.3 cm; Meyer, Rellingen, Germany) filled with 4,185 g of wet substrate. The water content of the substrate (determined gravimetrically) was 23% (based on the substrate wet mass; corresponding to 30% based on the dry mass of the substrate). To simulate field conditions with competition for light, water and nutrients, one seedling was placed in the centre and served as target plant, the other four seedlings were planted around this plant with distances of 6 cm to the central plant. Pots had holes at the bottom and were placed on dishes to allow draining but restrict water loss. The pots were placed in a block design in a glasshouse chamber (11°C, 75% r.h., light:dark 12:12 h; ambient sunlight supplemented with 400 W lamps, Philips HPI-T Plus; Philips, Amsterdam, Netherlands) and their positions were randomised once to twice a week. Five weeks after sowing, the temperature was increased to 14°C, the r.h. was 64% and the photoperiod was set to 14 h per day for 7 days. For the remaining time, plants were kept at 19°C and 58% r.h. at a photoperiod of 16 h light:8 h dark. Every other day, 15 randomly chosen pots were weighed, pot masses were averaged and all pots were filled with tap water to a soil water content of 23% (based on the wet mass of the substrate; corresponding to 30% based on the dry mass of the substrate). Each pot received 5 g and 3 g of a mineral fertiliser (Plantosan N-P$_2$O$_5$-K$_2$O 20-10-15, containing 6% MgO, 2% S and traces of B, Cu, Fe, Mn, Mo and Zn; Manna, Düsseldorf, Germany) at 32 days and 68 days after sowing, respectively. The fertiliser was applied directly before watering all pots.

### Irrigation treatments

We aimed to simulate either lower overall water availability (continuous drought) or extreme weather events (prolonged drought and sudden rain events, i.e. pulsed drought) in line with

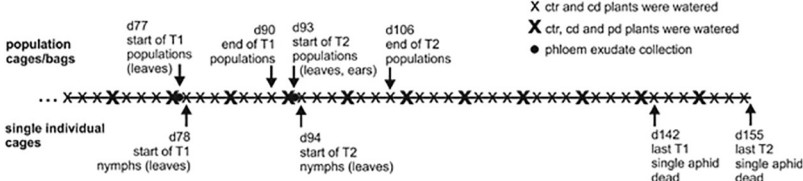

**Fig 1. Scheduling of conducted experiments.** Irrigation is shown from day 62 onwards when drought-stressed plants had dried to a soil water content of 11%; control (ctr) plants were watered to a soil water content of 23%, continuous drought-stressed (cd) plants were watered with 40% of the water ctr plants received, pulsed drought-stressed (pd) plants were watered with the cumulative amount of water cd plants received over 8 d. Aphid population assays are labelled above the line, single aphid assays are labelled below the line.

predicted current climate change scenarios [4], which crop plants likely face under field conditions. Therefore, different irrigation treatments were initiated 46 d after sowing, when stems started to elongate (BBCH stage 30 [34]). All pots were randomly assigned to one of three irrigation treatments, with 40 pots per treatment. Fifteen randomly chosen control (ctr) pots were weighed every other day and the mean amount of water needed to reach a soil water content of 23% was calculated. The respective amount of water was added to each pot of this treatment group. Pots that were subjected to the continuous drought (cd) or pulsed drought (pd) treatment were not watered until they reached an average soil water content of 11% (average of 15 randomly chosen pots). After this period (62 d after sowing until the end of the last aphid bioassays at day 155), pots of the cd treatment were watered every other day with 40% of the water amount that ctr plants received (Fig 1). Pots of the pd treatment received the cumulative amount of water that was given to cd plants only every eight days, with the first irrigation event at 68 d after sowing. Irrigation treatments were established in a pre-experiment in a way that drought-exposed plants showed signs of wilting but recovered under re-watering and had no signs of chlorosis or delay in development.

In a previous publication that focused on plant responses, we analysed the effects of the different drought stress regimes on plant biomass, physiology and the metabolome of entire flag leaves that had been harvested from one of the surrounding plants per pot [31]. For the present study, we used the central target plants from the same pots to collect phloem exudates and set up additional batches of plants for bioassays with aphids.

## Collection of phloem exudates

Phloem exudates were collected from the central target plants of the pots using the ethylenediaminetetraacetic acid (EDTA) method modified after Schweiger et al. [35] at two time points (T1, T2; Fig 1) during the experiment (n = 10 replicates per time point and treatment). At T1 (77 days after sowing and 31 days after start of the different irrigation treatments; start of plant heading, BBCH stage 51), exudates were collected from the three youngest fully developed leaves of the main shoot. Using another batch of pots, at T2 (93 days after sowing and 47 days after start of the different irrigation treatments; inflorescences fully developed, BBCH stage 59) exudates were collected from the three youngest fully developed leaves and in addition from the ear of the main shoot. These time points were chosen, because the transition from the vegetative to the flowering stage is crucial for the interaction of wheat with *S. avenae* aphids, which rapidly colonise emerging ears [36] and feed on the phloem sap of the inflorescences, mostly at or close to the inflorescence stem. Leaf blades were cut at the base close to the stem and pooled for each plant, whereas ears were cut 1 cm below the flowers. The following steps were performed in darkness at room temperature. Plant parts were placed with their cut surfaces in

50 ml tubes containing 1 ml of 8 mM EDTA solution (AppliChem, Darmstadt, Germany, pH = 7) for 2 h to suppress sieve tube plugging. Afterwards, the plant parts were transferred into new tubes containing 1 ml of Millipore water for 2 h to collect the phloem exudates. For blank sampling, tubes were filled with 1 ml of Millipore water and treated the same way as samples were treated. Samples and blanks were then frozen in liquid nitrogen and stored at -80°C. Plant material used for exudate collection was dried for 96 h at 40°C and dry mass was determined.

## Amino acid analysis

Amino acids in phloem exudates were analysed modified after Jakobs and Müller [37]. Subsamples (300 μl) of the phloem exudates were lyophilised and extracted in 60 μl of 80% methanol (LC-MS grade, VWR International, Leuven, Belgium) containing the internal standards norvaline and sarcosine (each at 50 pmol μl$^{-1}$, Agilent Technologies, Waldbronn, Germany). Samples were analysed via high performance liquid chromatography coupled to fluorescence detection (1260/1290 Infinity, Agilent Technologies, Santa Clara, CA, USA) using a ZORBAX Eclipse Plus C18 column (250 x 4.6 mm i.d., 5 μm, Agilent Technologies) with a guard column. Derivatisation of samples with borate buffer (Agilent Technologies, 0.4 M, pH = 10.2), ortho-phthaldialdehyde (Agilent Technologies, 10 mg ml$^{-1}$ in borate buffer and 3-mercaptoproprionic acid), 9-fluorenyl-methyl chloroformate (Agilent Technologies, 2.5 mg ml$^{-1}$ in acetonitrile) and injection diluent [100 ml mobile phase A (see below) and 0.4 ml 85% phosphoric acid (AppliChem)] was performed in the autosampler (6°C). The mobile phase A consisted of 1.4 g $Na_2HPO_4$ (AppliChem), 3.8 g $Na_2B_4O_7 \cdot 10 H_2O$ (Sigma-Aldrich, Steinheim, Germany) and 32 mg $NaN_3$ (Roth, Karlsruhe, Germany) in 1 l Millipore water (pH was adjusted to 8.2 and the eluent filtered through a 0.45 μm membrane). The mobile phase B was a mixture of methanol, acetonitrile (LC-MS grade, VWR International) and Millipore water (4.5:4.5:1, v:v:v). The flow rate was 1.5 ml min$^{-1}$ at 40°C column temperature. At the beginning, 2% B were held for 0.84 min, followed by a ramp to 57% B (reached at 43.4 min) and by column cleaning and equilibration. The excitation and emission wavelengths of the detector were set to 340 nm and 450 nm for the first 32 min (primary amino acids) and to 260 nm and 325 nm for the remaining time (secondary amino acids), respectively. Amino acids were identified by comparing their retention times with those of reference standards measured within the same worklist and quantified via peak areas using OpenLab ChemStation revision C.01.07 (Agilent Technologies). Peak areas of amino acids were divided by the peak areas of the internal standards (norvaline for primary amino acids, sarcosine for secondary amino acids) and by the corresponding dry mass of the plant material that had been used for the phloem exudate collection.

## Analysis of organic acids and sugars

Organic acid and sugar analysis was modified according to Kutyniok and Müller [38]. Subsamples (300 μl) of the phloem exudates were lyophilised and redissolved in 80% methanol with ribitol (99%, Sigma-Aldrich) as internal standard. Subsamples of the supernatants were dried under nitrogen and subsequently derivatised with O-methylhydroxylamine hydrochloride (Thermo Fisher Scientific, Karlsruhe, Germany) in pyridine (99.9%, Sigma-Aldrich, 20 mg ml$^{-1}$) for 90 min at 37°C and with N-methyl-N-trimethylsilyl-trifluoroacetamide (Macherey-Nagel, Düren, Germany) for 30 min at 37°C. Samples were analysed via gas chromatography coupled to mass spectrometry (GC-MS 2010 Plus QP2020, Shimadzu, Kyoto, Japan) using a VF-5ms column (30 m × 0.25 mm i.d., 10 m guard column, Varian, Palo Alto, CA, USA). Samples (1 μl) were injected at 225°C, using a split ratio of 1:10. The flow of the

carrier gas (helium) was set to 1.14 ml min$^{-1}$. The temperature gradient started at 80˚C that were kept for 3 min, followed by a ramp of 5˚C min$^{-1}$ to 310˚C. Ions from 40 to 600 m/z were detected in electron impact ionisation mode at 70 eV. Peaks were identified by comparing Kováts retention indices based on *n*-alkanes (C8-C40, Sigma-Aldrich) as well as mass spectra to reference substances measured under the same conditions and to an in-house database. Analytes were quantified as peak areas via the total ion count. Where required (i.e., for fructose and glucose), peak areas of analytes belonging to the same metabolite were summed up. Peak areas were divided by the peak area of the internal standard and by the corresponding dry mass of the plant part, from which the phloem exudates had been collected.

## Aphid bioassays

Aphids of *Sitobion avenae* were obtained from Koppert Biological Systems (Suffolk, UK) and reared in the laboratory on *T. aestivum* (cv. Tybalt) plants (2–5 weeks old) in insect rearing tents (60 x 60 x 60 cm) for several generations. Bioassays were initiated in parallel to the two phloem exudate collections (T1, T2; Fig 1), using separate plant batches (n = 10 replicates per treatment and time point). The population dynamics of groups of five adult aphids as well as the survival of individual nymphs were monitored on the central target plants of each pot by confining aphids in different cages/bags. Tests were performed at two time points, with younger (T1) and older (T2) plants. To assess the population dynamics, five apterous adult aphids (= 'aphid population') were placed in clip cages (inner diameter 16 mm, height 15 mm) on the second youngest fully developed leaf of the main shoot (~7 cm away from the stem) of each target plant. On plants used at T2, additionally five apterous adult aphids were placed on the ear of the main shoot of the target plant, confined in gauze bags (length 18 cm, diameter 6.5 cm, mesh size 210 μm). On the following day, one nymph that had been born in the meantime by each aphid population on the leaves was placed on the youngest fully developed leaf of the same plant in a separate clip cage to monitor its survival (see below). All other nymphs, which had been born within this first day on the leaves and on the ears, were discarded. From the second day on, all offspring was left in the cages/bags and all individuals per population on the leaves and ears were counted every other day until day 13 after initiating the population assays. Population assays were terminated then, because the cages /bags became very crowded (> 100 aphids on some plants).

To score the survival of individual aphids placed in the separate cages on leaves, their condition (alive or dead) was assessed daily until all aphids were dead (T1: 64 d, T2: 61 d after nymphs had been put into the individual cages). Survival was only scored on the leaves but not on the ears, because multiple cages could only be clipped on leaves. Offspring delivered later by these individual aphids was discarded daily to avoid competition.

## Data processing and statistical analyses

Only those metabolites (amino acids, organic acids, sugars) were retained in the data set which did not occur in the blanks or for which the concentrations in the samples were much higher than in the blanks. Moreover, only samples, in which at least 4 metabolites were detected in the GC-MS analysis, were included for further data processing, resulting in 5–10 replicates per treatment group. Metabolites had to occur in at least 50% of the replicates of at least one of the treatment groups to be retained in the data set (in accordance with other metabolomics studies, e.g. [31, 32]). To compare the phloem exudate composition in dependence of the irrigation treatment, relative abundances of amino acids, organic acids and sugars in percent (sum of all three compound classes set to 100%) were used for leaves and ears. We did not use absolute

concentrations, because the phloem exudation rate during collection may differ between plants of different irrigation treatments and between plant parts.

If not stated otherwise, further analyses were performed in R 3.4.2 [39], using the indicated packages. Nonmetric multidimensional scaling plots were generated for relative abundances of amino acids (sum set to 100%) and organic acids and sugars (sum set to 100%) across as well as within time points and plant parts, using Wisconsin double standardisations of square root-transformed data and Kulczynski distances (package 'vegan' [40]).

For each metabolite, the $log_2$ fold change (mean relative abundance in the treatment divided by the mean relative abundance in the respective control group) was calculated. To compare the phloem exudate composition of leaves and ears, additional $log_2$ fold changes were calculated (mean relative abundance in ears at T2 divided by mean relative abundance in leaves at T2) within each irrigation treatment. These fold changes were used to generate average linkage hierarchical cluster heatmaps based on Euclidean distances with Cluster 3.0 [41] and Java TreeView [42]. Only metabolites were included that occurred in at least 50% of the replicates of at least one of the two groups that were compared. Metabolites were clustered, but not treatment groups.

Differences in aphid population sizes after 13 days were tested using a generalised linear model with the factors irrigation treatment (levels ctr, cd, pd) and plant part (levels T1 leaves, T2 leaves, T2 ears) as well as their interaction with a quasi-Poisson distribution and log link function. Additionally, for each time point and plant part manual contrasts were calculated for ctr plants vs. cd plants and for cd plants vs. pd plants with the package 'contrast' [43] and P-values were corrected according to Holm within each time point and plant part. Kaplan-Meier survival probability curves (R package 'survival' [44]) were plotted for individual aphids on leaves, accounting for right censoring for few aphids (7 out of 60) that escaped from the cages. To test the effects of the factors irrigation and time point (levels T1, T2) as well as their interaction on the survival of individual aphids on leaves, a linear Cox model was performed (R package 'coxme' [45]), followed by pairwise log rank tests (R package 'survival' [44]) with Holm correction of P-values within each time point to compare the aphid survival on ctr vs. cd plants as well as on cd vs. pd plants.

## Results

### Metabolite composition of phloem exudates

In total, 20 amino acids (Figs 2A and 3), 5 organic acids and 3 sugars (Figs 2B and 3) were detected in the phloem exudates. The cis- and trans-isomers of aconitic acid could not be distinguished, i.e., the corresponding peak may be one of the isomers or the sum of both. The primary metabolite composition of the phloem exudates was partly influenced by the irrigation treatments, time points and plant parts (Fig 2). The amino acid profiles of phloem exudates of ctr leaves and cd leaves at T1 overlapped largely but ctr leaves were separated from pd leaves (Fig 2C), whereas those of ctr leaves at T2 were separated from those of both drought groups (Fig 2E). Furthermore, at T2 the amino acid profiles of phloem exudates of ears and leaves were quite distinct (Fig 2A). Within phloem exudates of ears, especially the pd group differed from the ctr group (Fig 2G). For organic acids and sugars, the phloem exudates of T1 leaves partly differed from those of T2 leaves and the exudate composition of ears was clearly separated from that of leaves (Fig 2B). The irrigation treatments did not cause any clear separation regarding organic acids and sugars in the phloem exudates of the different plant parts collected at T1 and T2 (Fig 2D, 2F and 2H).

The cluster heatmap (Fig 3A) revealed that the responses of proline, asparagine and aconitic acid to the drought treatments were most pronounced. The relative concentrations of many

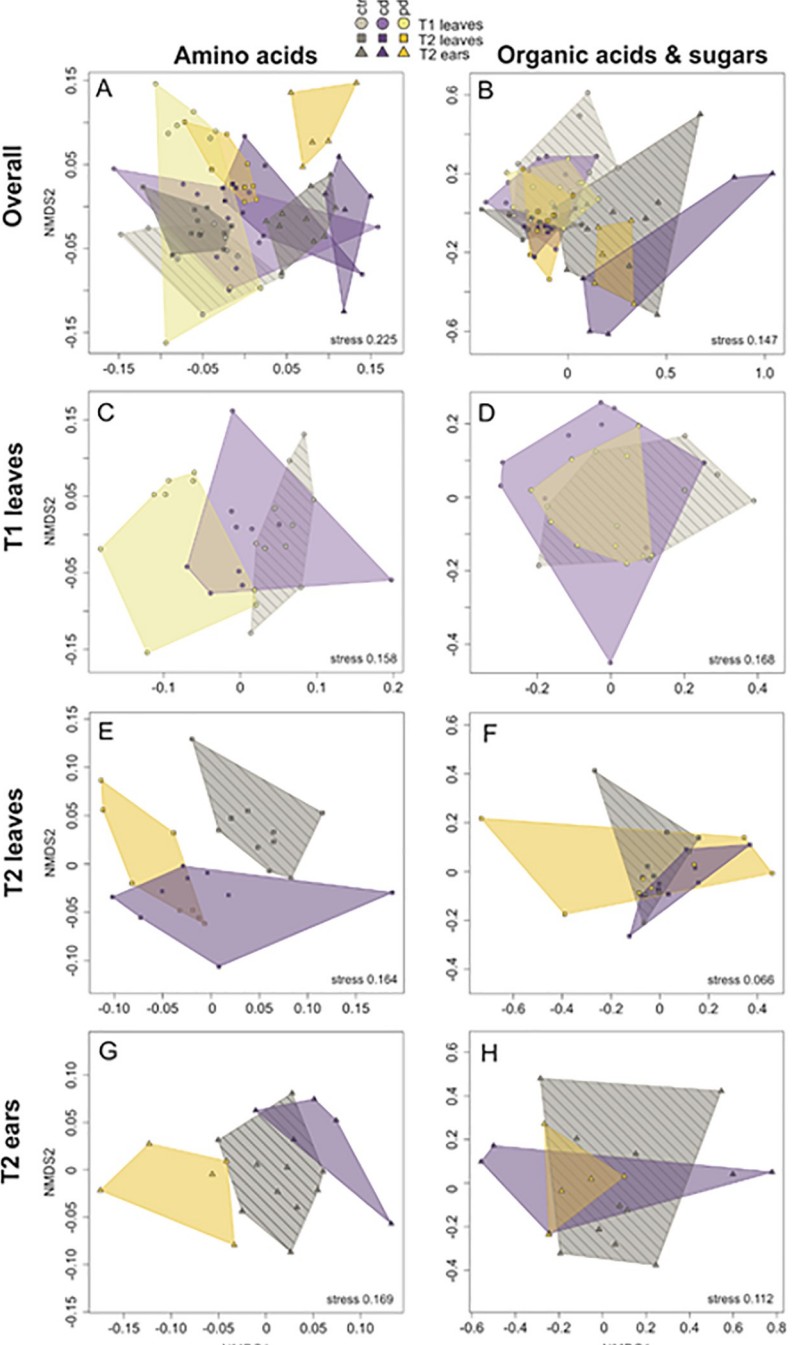

**Fig 2. Nonmetric multidimensional scaling plots of amino acids (left) and organic acids with sugars (right) detected in phloem exudates of *Triticum aestivum* measured at two different time points after the beginning of the irrigation treatments (T1 = 77 days, T2 = 93 days after sowing) in different plant parts (leaves, ears).** Cd (continuously drought-stressed) and pd (pulsed drought-stressed) plants were compared to well-watered control (ctr) plants. NMDS plots for phloem exudates over all time points and plant parts (**A**, **B**), for leaves at T1 (**C**, **D**), for leaves at T2 (**E**, **F**) and for ears at T2 (**G**, **H**). Analyses were based on relative abundances of the metabolites. Stress values are given at the bottom. The treatment groups are framed by convex hulls, with the corresponding areas being hatched (ctr) and non-hatched (drought-stressed plants), respectively. n = 5–10 replicates per treatment.

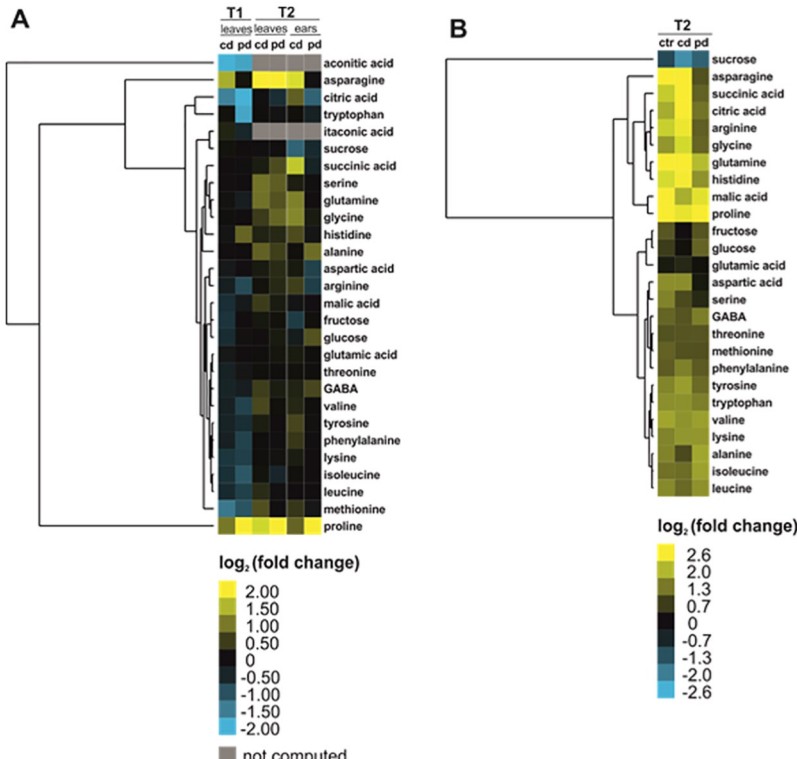

**Fig 3. Average linkage hierarchical cluster heatmaps (based on Euclidean distances of log$_2$ fold changes) of relative metabolite abundances in phloem exudates of leaves and ears of *Triticum aestivum* measured at two time points after the beginning of the irrigation treatments (T1 = 77 days, T2 = 93 days after sowing).** In **A**, cd (continuously drought-stressed) and pd (pulsed drought-stressed) plants were compared to well-watered control plants for each time point and plant part. In **B**, phloem exudates of ears were compared to those of leaves within each irrigation treatment at T2. For some comparisons, no fold change was computed since the metabolite did not occur in at least 50% of the replicates of at least one of the two groups that were compared (grey boxes). GABA, γ-aminobutyric acid. Means of n = 5–10 replicates per treatment.

amino acids were lower in leaf phloem exudates of drought-stressed compared to control plants at T1 but higher or comparable in phloem exudates of leaves and ears of drought-stressed plants collected at T2. The strongest changes in relative concentrations were found for proline, which had up to 13-, nine- and six-fold higher mean relative concentrations in phloem exudates of leaves at T1, leaves at T2 and ears (T2) of pd plants compared to control plants (Fig 3A). In contrast, asparagine showed a seven-fold higher relative concentration in leaf phloem exudates of cd compared to control plants at T2. Aconitic acid had considerably lower relative concentrations in leaf phloem exudates of drought-stressed plants than in those of ctr plants at T1; at T2, it only occurred in few replicates and no fold changes were computed. Similarly, the relative citric acid concentrations were markedly lower in leaf phloem exudates of drought-stressed (especially pd) plants compared to those of controls at T1. For the sugars we found no clear pattern except for sucrose that showed 2.3 and 1.5 times higher relative concentrations in phloem exudates of ctr ears (T2) than in those of cd and pd ears, respectively. Comparing phloem exudates of ctr ears versus leaves at T2, the mean relative concentrations of asparagine, proline and glutamine as well as of malic acid were 15, 10, 6 and 6 times higher in ears (Fig 3B). Similar differences were also observed in phloem exudates of ears versus leaves of both drought stress treatments. In contrast, we found two times lower relative sucrose

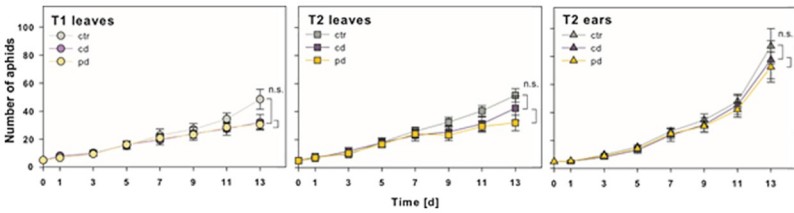

**Fig 4. Mean numbers (± standard errors) of *Sitobion avenae* aphids (nymphs and adults) on *Triticum aestivum* leaves and ears over time.** Aphids were placed on control (ctr), continuously drought-stressed (cd) or pulsed drought-stressed (pd) plants at two time points after the beginning of the irrigation treatments (start at T1: 77 days after sowing or T2: 93 days after sowing). Bioassays started with 5 adult aphids per clip cage (leaves) or gauze bag (ears). For each time point and plant part at day 13, manual contrasts of ctr vs. cd and cd vs. pd were calculated; n.s. = not significant; n = 10 replicates per treatment.

concentrations in the phloem exudates of ears than in those of leaves (ctr, T2). These differences were even more pronounced in drought-stressed plants (Fig 3B).

## Aphid population growth and survival

Aphid population sizes increased from five up to 162 aphids within 13 days (Fig 4). At day 13 of the bioassay, population sizes were significantly influenced by the factors irrigation and plant part (Table 1), being on average between 13% and 61% larger on ctr plants compared to cd and pd plants, respectively. This trend could be observed from day 7 onwards (Fig 4). However, population sizes at day 13 did not differ significantly between ctr and cd plants or between cd and pd plants. Moreover, on average two times larger populations were found at T2 on ears compared to leaves across plants of all irrigation treatments at day 13 (Fig 4).

The survival of individual aphids on leaves was significantly influenced by the irrigation treatment (Table 2). Aphids survived longest on ctr plants and showed the lowest survival probability on pd plants at T1 (Fig 5). The effects of the drought treatments were slightly more pronounced at T1 but did not differ significantly between T1 and T2, i.e. there was no significant effect of time point and also no significant interaction between irrigation and time point (Table 2). For the bioassay starting at T1, aphid survival was significantly lower on cd compared to ctr plants, whereas the survival probability did not differ significantly between aphids on cd and pd plants at T1 or between the treatment groups (ctr vs. cd, cd vs. pd) at T2.

## Discussion

Effects of drought stress on the metabolite profiles of phloem sap may affect plant-aphid interactions, because aphids use the phloem sap as their food source. Our study revealed that

**Table 1. Output of generalised linear model for population sizes of *Sitobion avenae* aphids on *Triticum aestivum* subjected to different irrigation treatments (factor irrigation with levels well-watered, continuous drought, pulsed drought).**

|  | Residual deviance | Residual df | *F* | *P* |
|---|---|---|---|---|
| Null Model | 1717.0 | 89 | - | - |
| Irrigation | 1627.6 | 87 | 3.56 | **0.033** |
| Plant Part | 1056.5 | 85 | 22.74 | **< 0.001** |
| Irrigation x Plant Part | 1033.6 | 81 | 0.46 | 0.768 |

At two time points after the beginning of the irrigation treatments (T1: 77 days after sowing or T2: 93 days after sowing), aphid populations were placed on different plant parts (factor plant part with levels T1 leaves, T2 leaves, T2 ears). Final aphid population sizes after 13 days were tested. Significant *P*-values (< 0.05) are highlighted in bold; n = 10 replicates per treatment.

**Table 2. Output of linear Cox model for the survival probability of *Sitobion avenae* aphids on *Triticum aestivum* subjected to different irrigation treatments (factor irrigation with levels well-watered, continuous drought, pulsed drought).**

|  | df | $X^2$ | P |
|---|---|---|---|
| Irrigation | 2 | 11.07 | **0.004** |
| Time point | 1 | 0.82 | 0.366 |
| Irrigation x Time point | 2 | 4.26 | 0.119 |

Individual nymphs were placed on leaves at two different time points after the beginning of the irrigation treatments (start at T1: 78 days after sowing or T2: 94 days after sowing). Significant *P*-value ($< 0.05$) is highlighted in bold; n = 10 replicates per treatment.

drought stress caused changes in the metabolic composition of the phloem exudates of both leaves and ears of wheat, with fold changes of some metabolites being higher in pd compared to cd plants, as predicted. In particular, proline was drastically enhanced in relative abundance in response to drought stress. Proline is well known to accumulate under drought conditions in leaves, as it acts as an osmolyte and is involved in protecting cell structures and functions [46]. Enhanced proline concentrations in leaves under drought are often due to an interplay between increased biosynthesis and reduced catabolism [47] and proline may be transported from photosynthetic tissues to roots under drought [48]. Proline has also been found to be enhanced in leaf phloem sap or exudates of drought-exposed plants of several species, including wheat [24, 32, 49]. The present study revealed a drought-induced increase of the relative concentrations of proline in phloem exudates not only of leaves but also of ears and an overall much higher relative concentration in the latter. Similar to proline, asparagine increased in phloem exudates of drought-exposed plants and was particularly high in exudates of ears. Asparagine is likewise responsive to abiotic stress [50] and was found to increase in concentration under drought stress in leaves and their phloem exudates in different species [26, 51, 52]. Whether the relative increases of proline and asparagine in the phloem exudates under drought found in the present study are due to lower protein biosynthesis, higher degradation of proteins or to *de novo* synthesis or lower catabolism of these amino acids remains to be tested.

Interestingly, some metabolites in the phloem exudates mainly increased relatively in response to the cd treatment (e.g., asparagine), others in response to the pd treatment (e.g., proline). The latter may be explained by longer drought stress periods that led to intermittently lower soil water contents. Some metabolic processes might only become modulated above a certain stress threshold. In contrast, a more pronounced impact on phloem metabolite concentrations caused by cd stress might be explained by the fact that cd plants only received a fraction of the water that pd plants received on the day before the phloem exudate collection. Thus, cd plants might have had a lower chance to recover from drought compared to pd plants. Moreover, strikingly many amino acids showed lower relative concentrations in phloem exudates of leaves in drought-stressed compared to control plants at T1, while these were similar or higher compared to the ctr plants at T2. These effects might be caused by the varying developmental stages of the plants at harvest and/or the different durations of the drought stress experienced before. After short-term stress these metabolites may be used to synthesise other compounds that are involved in the drought response. After a longer period of stress these amino acids might be synthesised in response to drought.

With regard to organic acids, the most apparent decreases in relative concentrations in response to drought were found for citric acid and aconitic acid in phloem exudates of leaves at T1. Likewise, a decrease of citric acid was found in shoots of two genotypes of wheat

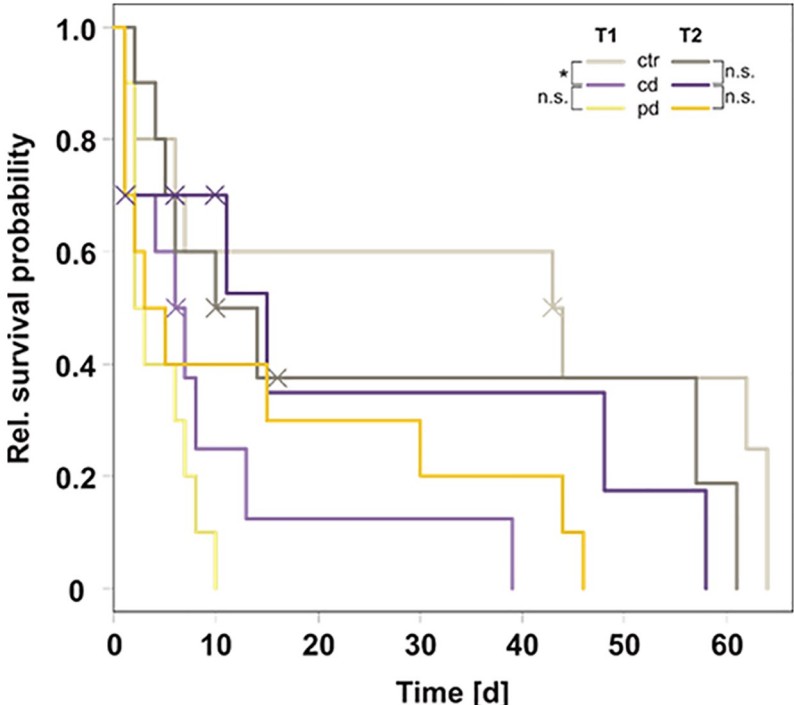

**Fig 5. Kaplan-Meier relative survival probability curves of *Sitobion avenae* aphids on *Triticum aestivum* leaves over time.** Nymphs were placed on control (ctr), continuously drought-stressed (cd) or pulsed drought-stressed (pd) plants at two different time points after the beginning of the irrigation treatments (start at T1: 78 days after sowing or T2: 94 days after sowing). Crosses indicate right censoring. For each time point, pairwise log rank tests comparing the groups ctr vs. cd and cd vs. pd were calculated; n.s. = not significant, * = $P < 0.05$; n = 10 replicates per treatment.

seedlings after 15 d of water deprivation [53]. Citric acid is an intermediate of the tricarboxylic acid cycle; thus, the energy metabolism is apparently affected by drought. Impacts on the energy metabolism might not only induce further metabolic changes but also affect plant physiology and productivity [54]. For aconitic acid, we could not distinguish the isomers (*cis/trans*). As *cis*-aconitic acid is also part of the tricarboxylic acid cycle, a decrease of this organic acid would again point to changes in the energy metabolism. However, grasses including wheat also contain *trans*-aconitic acid [55]. Interestingly, this organic acid showed lower concentrations under salt stress in shoots of *Zea mays* (Poaceae) [56] and led to reduced survival of the aphid *Rhopalosiphum padi* (Hemiptera: Aphididae) on an artificial diet [57]. Furthermore, a negative correlation between the concentration of *trans*-aconitic acid in the stem juice of different sorghum (Poaceae) cultivars and damage by the sugarcane aphid has been reported [58], highlighting that organic acids can impact aphid development.

For sugars, no strong responses to drought were found in phloem exudates of wheat leaves in the present study. Sugars are known to contribute to drought stress tolerance; however, an increase under drought seems to occur only in certain genotypes of wheat [53, 59, 60]. In leaf phloem exudates of drought-stressed plants of *Arabidopsis thaliana* (Brassicaceae) sucrose likewise accumulated compared to control plants [25]. In contrast, in phloem exudates of ears examined in our study, the relative sucrose concentration was lower in drought-stressed compared to ctr plants. In ears, relative concentrations of metabolites in phloem sap might be influenced by an altered metabolism during grain loading. Metabolic changes over time (from anthesis to grain filling) were shown for different organs of wheat and depended on the water

status of the plants [61]. Our wheat plants might have been in different developmental stages of anthesis or grain loading due to different watering regimes at T2, although their habitus did not vary visually. Sucrose is an abundant sugar in pure phloem sap of all plant species, whereas the occurrence of hexoses such as glucose and fructose in the phloem sap is still a matter of debate [62] and is often attributed to artefacts imposed by the EDTA-facilitated exudation method [63]. However, in wheat phloem sap collected via aphid stylectomy close to the ear glucose and fructose were also detected and their levels increased significantly during grain loading [64]. Thus, the hexoses in the phloem exudates found in the present study in wheat are probably not artefacts from the sampling method.

In response to the drought-induced changes in phloem quality, we hypothesised that aphids show a higher population growth and survival on drought-stressed compared to control plants in line with the plant stress hypothesis [16]. However, while there was an overall significant effect of irrigation, within time points and plant parts the population sizes of *S. avenae* did not differ significantly between cd and ctr plants. In contrast, the relative survival probability was significantly higher on ctr plants (at T1). Because the control plants showed the highest shoot biomass [31], these results rather support the plant vigour hypothesis [15]. In a recent study, a lower net reproductive rate and reduced rates of increase on drought-stressed compared to control wheat plants for *S. avenae* were reported [60]. In our study, the changes in metabolite composition of the phloem exudates induced by the different irrigation treatments may not have been substantial enough to impact overall population size, although they affected the individual survival (at T1).

Moreover, in contrast to our expectation and to the pulsed stress hypothesis [17], the aphid population growth and survival did not differ significantly between cd and pd plants, although some phloem metabolites such as proline and citric acid were more affected by pd than by cd. Due to potential effects of the irrigation treatment on the volume of collectable phloem exudates, evaluating the absolute concentrations of amino acids, organic acids and sugars was not possible. Collection of phloem sap via aphid stylectomy from differently watered plants in a similar experimental set-up led to considerable higher volumes collected per time obtained from ctr plants compared to cd and pd plants (Stallmann, personal observation). This might be explained by an increase in phloem sap viscosity under drought and a resulting reduction in its flow rate [65]. The plant water status and leaf turgor are decisive for the accessibility of phloem sap and drought might therefore influence the survival of *S. avenae* on drought-stressed wheat plants, probably by compromising aphid feeding. In line with that assumption, the phloem sap ingestion rate of *R. padi* was reduced on drought-stressed *Dactylis glomerata* (Poaceae) plants despite an enhanced osmotic pressure and slightly higher concentrations of essential amino acids in the phloem sap, leading to a suppressed intrinsic rate of increase [27].

Water availability does not only influence primary metabolites in plant tissues but can also affect specialised (secondary) metabolites such as terpenoids or alkaloids, which increase in certain plant parts under drought conditions depending on the duration of stress [12, 66, 67]. Some of the characteristic defence metabolites of wheat, the benzoxazinoids, are known to accumulate under drought conditions in leaves of seedlings [68, 69] and likewise in flag leaves of mature plants under pd stress [31]. Benzoxazinoids also occur in the phloem sap of wheat [70] and may affect aphid development [70, 71]. Other abiotic factors related to climate change such as warming have been shown to modulate the salicylic and jasmonic acid signalling pathways in wheat, with effects on the population size and feeding behaviour of *S. avenae* [72]. Furthermore, drought can impact morphological plant parameters [10], such as the trichome density, which can influence aphid survival [73].

Aphid populations grew almost twice as large on ears compared to leaves on older plants (T2) in the present experiment. In the field, *S. avenae* cause highest yield losses when

proliferating on the inflorescences during flower formation and anthesis [30] and many aphids colonise the ears as soon as they emerge, while some mostly feed on the phloem sap of the upper leaves ([36]; personal observation). That these aphids prefer the ears over the leaves may be related to better nutritional conditions at the ears, which may also explain the higher population growth on ears in our study. Important nutrients are transported from the vegetative tissue to the developing grains via the sieve tubes, leading to changes in phloem sap composition during grain loading [64]. In our experiment, the relative concentrations in phloem exudates collected from ears of ctr plants were lower for sucrose but higher for malic acid and several amino acids like proline, asparagine and glutamine compared to leaf phloem exudates. Since both sucrose and amino acids affect aphid development [74], these differences may in part explain the larger aphid populations on ears than on leaves.

## Conclusion

In summary, our study revealed that drought influences the relative composition of phloem exudates of wheat plants with the magnitude and direction of the impacts depending on the irrigation frequency, time point and plant part under investigation. Regarding the two time points, differences in plant phenology as well as differences in the duration of the preceding drought stress phase may have contributed to the changes in plant quality. These modifications of the phloem sap composition might be one of the reasons for changes in aphid survival on plants subjected to drought stress. Considering these results in context of the predicted scenarios of climate change, future investigations on drought and other components of climate change should take into account that not only the intensity but also the frequency of stresses can be decisive for plant chemistry and herbivore development. In addition to the severity of drought stress, the time at which the drought stress occurs during plant development may play a role for wheat grain production, as shown before [75]. Further studies are needed to investigate the underlying mechanisms.

## Supporting information

**S1 Data.**
(XLSX)

## Author Contributions

**Conceptualization:** Jana Stallmann, Rabea Schweiger, Caroline Müller.

**Data curation:** Jana Stallmann.

**Formal analysis:** Jana Stallmann.

**Funding acquisition:** Caroline Müller.

**Investigation:** Jana Stallmann, Caroline A. A. Pons.

**Methodology:** Rabea Schweiger.

**Project administration:** Caroline Müller.

**Resources:** Caroline Müller.

**Supervision:** Caroline Müller.

**Validation:** Rabea Schweiger, Caroline Müller.

**Visualization:** Jana Stallmann.

**Writing – original draft:** Jana Stallmann.

**Writing – review & editing:** Caroline A. A. Pons, Rabea Schweiger, Caroline Müller.

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
