## [Decision Letter · Decision Letter 0]

9 Nov 2021

PONE-D-21-29624Time point- and plant part-specific changes in phloem exudate metabolites of leaves and ears of wheat in response to drought and effects on aphidsPLOS ONE

Dear Dr. Müller,

Thank you for submitting your manuscript to PLOS ONE. After careful consideration, we feel that it has merit but does not fully meet PLOS ONE’s publication criteria as it currently stands. Therefore, we invite you to submit a revised version of the manuscript that addresses the points raised during the review process by the 3 reviewers and the editor (below). Please ensure that your decision is justified on PLOS ONE’s publication criteria and not, for example, on novelty or perceived impact.

We look forward to receiving your revised manuscript.

Kind regards,

Nicolas Desneux

Academic Editor

PLOS ONE

Journal Requirements: 

Editor Comments:

In addition to the comments from the 3 reviewers (below):

- The authors may consider relying on the recent key review article by Han et al. (in Annu Rev Entomol) about importance of bottom-up forces on pests in agro-ecosystems.

Han P, et al. Bottom-Up Forces in Agroecosystems and Their Potential Impact on Arthropod Pest Management. Annual Review of Entomology. https://doi.org/10.1146/annurev-ento-060121-060505.

- Please avoid referring to grey literature papers when WoS indexed papers are available. For example grey literature references 26 and 27 (in German) could be omitted and the citation by Hullé et al. 2020 (linking to the Encyclop'Aphid) could be cited here (this citation also covering facts reported in references 28 and 29).

* Hulle M, Chaubet B, Turpeau E, Simon JC. 2020. Encyclop’Aphid: a website on aphids and their natural enemies. Entomol. Gen. 40:97–101

Reviewers' comments:

Reviewer's Responses to Questions

**Comments to the Author**

1. Is the manuscript technically sound, and do the data support the conclusions?

Reviewer #1: Yes

Reviewer #2: Yes

Reviewer #3: Yes

2. Has the statistical analysis been performed appropriately and rigorously? 

Reviewer #1: Yes

Reviewer #2: Yes

Reviewer #3: Yes

3. Have the authors made all data underlying the findings in their manuscript fully available?

Reviewer #1: No

Reviewer #2: Yes

Reviewer #3: Yes

4. Is the manuscript presented in an intelligible fashion and written in standard English?

Reviewer #1: Yes

Reviewer #2: Yes

Reviewer #3: Yes

5. Review Comments to the Author

Reviewer #1: The Introduction does a reasonable job of summarizing the previous literature on the effects of drought stress on insects, with a focus on aphids. As noted by the authors, performance of aphids on plants under drought stress can be affected negatively, positively, or not at all. This study adds another data point to this literature and is valuable for that reason. It would be nice, however, to see research that would elucidate general principles that allow us to predict aphid performance.

The experiment in general, and the drought stress treatments in particular, seem to have been carried out carefully and thoughtfully. Drought stress was defined solely by irrigation regime. “Stress” experienced by plants here was not rooted or grounded in any “objective” measure of plant physiological stress, such as turgor pressure or plant growth or photosynthetic rate. Furthermore, the authors do not provide descriptions of the conditions of the plants under the various irrigation regimes – i.e., were plants in the cd treatment wilted or chlorotic? Was growth reduced or growth patterns altered? For this reason, it is difficult to compare the stress experienced by plants in this study with stress experienced by plants in other studies, which in turn hinders our ability to draw general principles from specific studies. It would also have been nice for the authors to relate the stress imposed here with stress typically experienced by wheat under field conditions.

I was a bit disappointed by the Discussion because of its lack of specificity in relating biochemical changes with changes in aphid performance. It seems to me that the potential of the cluster analyses for illuminating the nature of the biochemical differences between treatments was not put to good use in the Discussion, which really provides little insight into the reasons for differences in performance on different plant parts/ages.

Lines 96-112 – I think this paragraph should incorporate more specific information from the authors’ previous study in the system – this information could have been used to generate more specific hypotheses

Lines 100-101 – T1 and T2 in this context are meaningless to readers – can you substitute actual time points?

Line 111 – instead of “fruits”, “kernels”? “grains”?

Line 156-160 – As noted above, I would like to have seen a short summary of the effects of these irrigation regimes on plant physiology – if not here, then in the Introduction. This perhaps would have formed the basis of more specific hypotheses in the Introduction

Lines 172-173 – the authors really should tell the readers here or elsewhere where, specifically, the aphids feed. For example, do they feed on developing kernels, inflorescences, petioles? Do the sites for phloem collection match the feeding sites? This is crucial information.

Line 255 – the writing is a little confusing here. Was this nymph produced by one of the five apterous adult aphids placed previously – or is this a separate placement? Also, in line 263, are the authors saying that some nymphs lived 61 – 64 days? After reading the Results, it is clear that the survival and population growth experiments were separate, but this was not communicated clearly in the Materials and Methods. In general, the entire section on aphid bioassays should be revised for clarity

Line 270-275 – the writing in this section is ambiguous and the criteria for retaining compounds in the analyses seems arbitrary. Perhaps include more details.

Lines 303 and following – I understand that this approach to analyzing metabolite concentrations is fashionable, but it does little to tell the reader what nutritional conditions are actually being experienced by aphids. I understand also the reasoning for expressing everything as “relative concentrations”, but it would be nice to know if, for example, aphids feeding on drought-stressed plants had access to more free amino acids.

Line 340, “markedly” rather than “pronouncedly”

Line 430-432 – I don’t see the relevance of this to the study

Line 490, were the well-watered plants visibly or measurably more “vigorous”? This information was not provided

Line 501, surely there has to be published information related to this point?

Line 507, delete “of”

Reviewer #2: In this study, the authors investigated the responses of spring wheat and the aphid Sitobion avenae to continuous or pulsed drought. Population growth and survival of the aphid or relative concentrations of metabolites in the phloem of different plant parts were monitored .The research topic in the manuscript is very interesting and it is useful for the development of the insect pest control. Through analyzing the effects of drought on the content of plant metabolites and the growth and reproduction of aphid under drought conditions, so as to provide a basis for the pest control caused by environmental changes in the future.Currently this study is deficient in several areas. I suggest give the MS minor revision.

The content of the whole study method design is insufficient. For example, in Aphid bioassays part , the authors only test the survival and population of the aphid, and not the aphid feeding behavior and other aphid performance. Because the aphid feeding behavior maybe is more related to the phloem composition. Especially the amino acid content. Besides, the drought or other environment change can induce the expression of defense signaling genes or proteins, which can also impact the aphid performance. Please refer to the reference: Effects of field simulated warming on feeding behavior of Sitobion avenae (Fabricius) and host defense systems，Entomologia Generalis, DOI: 10.1127/entomologia/2021/1271

The Key word “aphid performance” is not suitable here. the authors only test the survival and population of the aphid, however, the word “performance” should include more.

statistical analyses:Authors discussed in the conclusion that “these modifications of the phloem sap composition might be one of the reasons for changes in aphid performance on plants subjected to drought stress”. So, the authors should analyze that the relationship between the composition of the phloem sap and the aphid performance?Line 536-537

In introduction: There are also many related studies in this area, and the author should include these references here, such as Line 52，54

In method, The research method is too complicated to describe clearly. Especilly in the part of Irrigation treatments, the descriptions of definition of ctr, cd , and pd were unclear.for example: “randomly chosen control (ctr) pots were weighed every other day and the mean amount of water needed to reach a soil water content of 23% (30% based on dry mass)” it is means that the water amount of ctr treatment is equal to 23*30% *dry mass or 30%*dry mass? And what is dry mass? Please give a clearer description in the whole part.

Reviewer #3: This manuscript reports on the effect of continuous and pulsed drought on amino acid, sugar and organic acids composition of different organs of wheat plants at two time points. Furthermore, they investigated the effect on aphids feeding on the respective plants and plant organs.

The manuscript is well written and the experiment is generally well explained and interpreted.

I only have a few comments and suggestions:

Under drought stress, plants tend to increase the number of trichomes to reduce water loss. Additionally, trichomes are an efficient morphological defense against small herbivores moving on the plant surface, like aphids. I wonder why the authors did not report any observations on trichome density.

You analyzed the concentrations of organic acids but the manuscript contains no explanation why they are relevant for the aphids.

Introduction:

l. 104: why do you expect less population growth on stressed plants compared to well-watered plants when you explained in lines 70-74 that amino acid and sucrose concentrations may be increased?

Material & Methods:

How did you determine the irrigation treatments? Have there been any pre-experiments? Why did you choose a soil water content of 23 and 11%, respectively?

L. 156-160: I am quite surprised to see that you published data from the same experiment in two different articles. Usually, that should have been reported together shouldn’t it?

Your irrigation treatments are quite complex and not very easy to understand. Maybe it would help to add an explanatory figure? Also, I am wondering for how many days the treatments continued: l. 152: “after this period…”: for how long?

Results:

Figure 1 is quite difficult to read because of the large fraction of overlay. I do appreciate the value in presenting all the data together, as an overview. However, in the current version you cannot see much more than that the samples are all quite similar. I suggest to create three figures each for amino acids and for organic acids, in a similar way as figure 3: one for T1-leaves, one for T2-leaves, and one for T2-ears.

Also, I don’t think grey was the best choice of color for the control because in some parts, it is hard to distinguish it from the dark purple.

Figure 3: please indicate significant differences between the treatments. I suggest naming the y-axis “number of aphids”.

Discussion:

Your time points contain two factors that differ. While you do mention this at one point or the other in the manuscript, I think it might be helpful to summarize it (additionally?) in one place. Between T1 and T2, the duration of the drought treatments progresses (short/long term) alongside plant phenology (vegetative/reproductive stage). Hence, both could explain observed changes.

l.431: please add that the accumulation is problematic from a human perspective

l. 496-498: Yes. Furthermore, this may also be important for the aphid population dynamics. Considering that you only observed aphid development for 13 days and that the pulsed drought treatment was watered every 8 days, it might be very relevant to show on which of these 13 days watering events occurred. It may have influenced your results greatly!

l.511: “specialized metabolites”: that is quite vage. Do you mean secondary metabolites? Plant defense metabolites?

l.514: here you should introduce the abbreviation BXD

l. 518: do you have a reference for the detrimental effect of BXDs on aphids?

l. 521: what are milk stages? Please elucidate

6. PLOS authors have the option to publish the peer review history of their article (what does this mean?). If published, this will include your full peer review and any attached files.

Reviewer #1: No

Reviewer #2: No

Reviewer #3: **Yes: **Christine Becker

---

## [Author Response · Author response to Decision Letter 0]

20 Dec 2021

COMMENTS BY THE EDITOR AND REPLIES

- The authors may consider relying on the recent key review article by Han et al. (in Annu Rev Entomol) about importance of bottom-up forces on pests in agro-ecosystems.

Han P, et al. Bottom-Up Forces in Agroecosystems and Their Potential Impact on Arthropod Pest Management. Annual Review of Entomology. https://doi.org/10.1146/annurev-ento-060121-060505.

- Please avoid referring to grey literature papers when WoS indexed papers are available. For example grey literature references 26 and 27 (in German) could be omitted and the citation by Hullé et al. 2020 (linking to the Encyclop'Aphid) could be cited here (this citation also covering facts reported in references 28 and 29).

* Hulle M, Chaubet B, Turpeau E, Simon JC. 2020. Encyclop’Aphid: a website on aphids and their natural enemies. Entomol. Gen. 40:97–101

REPLY: We thank the editor for pointing out the recently published review article by Han et al., which is now cited in line 58. 

The study published by Hullé et al. is now cited in lines 87-89. We rephrased the corresponding text slightly to better fit to the new reference and substituted the studies 26-29 as suggested.

COMMENTS BY REVIEWER #1 AND REPLIES

Reviewer #1: The Introduction does a reasonable job of summarizing the previous literature on the effects of drought stress on insects, with a focus on aphids. As noted by the authors, performance of aphids on plants under drought stress can be affected negatively, positively, or not at all. This study adds another data point to this literature and is valuable for that reason. It would be nice, however, to see research that would elucidate general principles that allow us to predict aphid performance.

REPLY: Thank you for all the constructive feedback. Yes, it would be indeed nice if such general principles could be found. However, nature is not that systematic and we feel that such generalisations are probably impossible, as responses to environmental factors and plant-herbivore interactions turned out to be highly species-specific. We added a statement addressing this aspect in the introduction, lines 80-81.

The experiment in general, and the drought stress treatments in particular, seem to have been carried out carefully and thoughtfully. Drought stress was defined solely by irrigation regime. “Stress” experienced by plants here was not rooted or grounded in any “objective” measure of plant physiological stress, such as turgor pressure or plant growth or photosynthetic rate. 

Furthermore, the authors do not provide descriptions of the conditions of the plants under the various irrigation regimes – i.e., were plants in the cd treatment wilted or chlorotic? Was growth reduced or growth patterns altered? For this reason, it is difficult to compare the stress experienced by plants in this study with stress experienced by plants in other studies, which in turn hinders our ability to draw general principles from specific studies. It would also have been nice for the authors to relate the stress imposed here with stress typically experienced by wheat under field conditions.

REPLY: Plants that were drought-exposed grew less than control plants, indicating that the plants experienced stress. We added this important information in the introduction (lines 89-92). Moreover, we justified why we used these specific irrigation treatments in the method section (lines 157-160: “We aimed to simulate either lower overall water availability (continuous drought) or extreme weather events (prolonged drought and sudden rain events, i.e. pulsed drought) in line with predicted current climate change scenarios, which crop plants likely face under field conditions.” Furthermore, we describe the plant phenology in response to drought now in more detail in lines 172-175: “Irrigation treatments were established in a pre-experiment in a way that drought-exposed plants showed signs of wilting but recovered under re-watering and had no signs of chlorosis or delay in development.”

I was a bit disappointed by the Discussion because of its lack of specificity in relating biochemical changes with changes in aphid performance. It seems to me that the potential of the cluster analyses for illuminating the nature of the biochemical differences between treatments was not put to good use in the Discussion, which really provides little insight into the reasons for differences in performance on different plant parts/ages.

REPLY: We agree that it would be interesting to better understand the relationships between the observed metabolic changes and aphid population growth respective survival. However, we cannot directly correlate the metabolic data with aphid population growth or survival, because phloem exudate collection and aphid bioassays were done using different batches of plants. Thus, we decided to only carefully address such potential relationships. Because metabolic differences between plant parts (ears vs. leaves) were larger than differences between plants of different age (T1 leaves vs. T2 leaves), we refer to differences in the relative concentrations of sucrose and amino acids to address differences between the plant parts that may affect aphids (lines 603-606).

Lines 96-112 – I think this paragraph should incorporate more specific information from the authors’ previous study in the system – this information could have been used to generate more specific hypotheses

REPLY: In the revised manuscript we added more specific information about the results observed in the previous study in lines 90-92 and 99-100. Based on these results and in line with suggestions by reviewer 3, we modified now our hypothesis and expected a higher population growth and better survival on drought-stressed plants (lines 117-119), because we had found previously a higher relative proline concentration in leaf phloem exudates of drought-stressed wheat plants (now reported in line 99-100) and we refer to the plant stress hypothesis.

Lines 100-101 – T1 and T2 in this context are meaningless to readers – can you substitute actual time points?

REPLY: We substituted T1 and T2 with day 77 and day 93 (line 109).

Line 111 – instead of “fruits”, “kernels”? “grains”?

REPLY: We now changed the phrase to “from vegetative to reproductive tissues”, avoiding the term “grains” (line 123).

Line 156-160 – As noted above, I would like to have seen a short summary of the effects of these irrigation regimes on plant physiology – if not here, then in the Introduction. This perhaps would have formed the basis of more specific hypotheses in the Introduction

REPLY: We now address the effects on the physiology (line 92) and on proline in phloem exudates (lines 99-100) of the plants in the introduction and modified one hypothesis (see reply above). 

Lines 172-173 – the authors really should tell the readers here or elsewhere where, specifically, the aphids feed. For example, do they feed on developing kernels, inflorescences, petioles? Do the sites for phloem collection match the feeding sites? This is crucial information.

REPLY: The aphids feed on the phloem sap of the inflorescences, mostly at or close to the inflorescence stem. We added this information in line 88. Thus, the sampling of ear phloem exudates by cutting the inflorescences 1 cm below the flowers allowed us to assess the metabolic composition close to the feeding sites of the aphids.

#Line 255 – the writing is a little confusing here. Was this nymph produced by one of the five apterous adult aphids placed previously – or is this a separate placement? Also, in line 263, are the authors saying that some nymphs lived 61 – 64 days? After reading the Results, it is clear that the survival and population growth experiments were separate, but this was not communicated clearly in the Materials and Methods. In general, the entire section on aphid bioassays should be revised for clarity

REPLY: The aphid bioassay part was revised thoroughly to make the procedure of the experiments clearer to the reader (lines 276ff). Furthermore, Figure 1 was added to visualise the schedule of the experiments and the timing of the different bioassays with aphids (population growth, survival of single individuals). The survival assays ended after 64 days (T1) and 61 days (T2), respectively. After these 64 or 61 days all aphids, whose survival had been monitored individually, had died; this in now additionally shown in Figure 1.

Line 270-275 – the writing in this section is ambiguous and the criteria for retaining compounds in the analyses seems arbitrary. Perhaps include more details.

REPLY: These criteria for retaining compounds in the statistical analyses are used in many metabolomics studies. Here, we now cite two recent metabolomics publications, where these methods were also applied (line 312).

Lines 303 and following – I understand that this approach to analyzing metabolite concentrations is fashionable, but it does little to tell the reader what nutritional conditions are actually being experienced by aphids. I understand also the reasoning for expressing everything as “relative concentrations”, but it would be nice to know if, for example, aphids feeding on drought-stressed plants had access to more free amino acids.

REPLY: Plant phloem sap is the food source of S. avenae aphids, as stated in line 87 and added now in line 475. Therefore, by analysing phloem exudates we could at least approximately uncover the relative metabolic composition that aphids feeding on the phloem sap of these plants would face. We agree that it would be nice to know how absolute concentrations of metabolites in the phloem exudates differ between the irrigation treatments. Unfortunately, when sampling the phloem exudates via the EDTA method, we cannot determine the volume of phloem exudate entering the collection solution and it may differ between differently watered plants (as explained in the method section, lines 315-317). Therefore, we think it is more accurate to only report on relative concentrations.

Line 340, “markedly” rather than “pronouncedly”

REPLY: We replaced “pronouncedly” by “markedly” (line 394).

Line 430-432 – I don’t see the relevance of this to the study

REPLY: We agree that this is more relevant from a human perspective and deleted the sentence. 

Line 490, were the well-watered plants visibly or measurably more “vigorous”? This information was not provided

REPLY: In the revised manuscript we mention visible effects on plants in the introduction (lines 90ff) and state now here again that these plants had more biomass in the discussion (line 557). Moreover, we specified the irrigation treatments in the methods by referring to wilting as well as signs of chlorosis and delay in development (lines 173-175).

Line 501, surely there has to be published information related to this point?

REPLY: Exactly to this point we could not find any published information. Nevertheless, the observed effects might be caused by changes in phloem sap viscosity under drought. We now mention that aspect here and cite the study by Sevanto (2018). 

Line 507, delete “of”

REPLY: We deleted the word “of”.

COMMENTS BY REVIEWER #2 AND REPLIES

Reviewer #2: In this study, the authors investigated the responses of spring wheat and the aphid Sitobion avenae to continuous or pulsed drought. Population growth and survival of the aphid or relative concentrations of metabolites in the phloem of different plant parts were monitored .The research topic in the manuscript is very interesting and it is useful for the development of the insect pest control. Through analyzing the effects of drought on the content of plant metabolites and the growth and reproduction of aphid under drought conditions, so as to provide a basis for the pest control caused by environmental changes in the future.Currently this study is deficient in several areas. I suggest give the MS minor revision.

REPLY: Thank you for the helpful comments.

The content of the whole study method design is insufficient. For example, in Aphid bioassays part , the authors only test the survival and population of the aphid, and not the aphid feeding behavior and other aphid performance. Because the aphid feeding behavior maybe is more related to the phloem composition. Especially the amino acid content. 

REPLY: We decided to assess aphid population dynamics and aphid survival, as these traits are probably most relevant to judge the impact of aphids on the plant and the damage to the crop. We agree that aphid feeding behaviour is also a very relevant aspect that probably depends on the phloem sap quality and that can affect aphid development. We thus mention the effects of drought on phloem sap ingestion rate and intrinsic rate of increase in the discussion (lines 577-580). Moreover, in the revised version of the manuscript we avoided the term “aphid performance”, as requested below.

Besides, the drought or other environment change can induce the expression of defense signaling genes or proteins, which can also impact the aphid performance. Please refer to the reference: Effects of field simulated warming on feeding behavior of Sitobion avenae (Fabricius) and host defense systems，Entomologia Generalis, DOI: 10.1127/entomologia/2021/1271

REPLY: Thank you for pointing us to this interesting reference. We added this aspect in the discussion of our manuscript and cited the paper by Wang et al. (lines 588-591).

The Key word “aphid performance” is not suitable here. the authors only test the survival and population of the aphid, however, the word “performance” should include more.

REPLY: We replaced the key word by “aphid population growth” and “aphid survival”. Likewise, throughout the text, we exchanged “performance” by respective more suitable terms.

statistical analyses:Authors discussed in the conclusion that “these modifications of the phloem sap composition might be one of the reasons for changes in aphid performance on plants subjected to drought stress”. So, the authors should analyze that the relationship between the composition of the phloem sap and the aphid performance?Line 536-537

REPLY: Indeed, the relationship of phloem sap composition and aphid population growth and survival would be interesting to investigate. However, our study design does not allow to test for direct correlations, because aphid bioassays and phloem exudate collection were carried out on different sets of plants.

In introduction: There are also many related studies in this area, and the author should include these references here, such as Line 52，54

REPLY: We specified the information and added references to these statements (lines 53-55).

In method, The research method is too complicated to describe clearly. Especilly in the part of Irrigation treatments, the descriptions of definition of ctr, cd , and pd were unclear.for example: “randomly chosen control (ctr) pots were weighed every other day and the mean amount of water needed to reach a soil water content of 23% (30% based on dry mass)” it is means that the water amount of ctr treatment is equal to 23*30% *dry mass or 30%*dry mass? And what is dry mass? Please give a clearer description in the whole part.

REPLY: As suggested by Reviewer 3, we added a figure describing the timeline of the experiments to make the methods clearer (Fig 1). We also thoroughly revised the method section for clarification.

COMMENTS BY REVIEWER #3 AND REPLIES

Reviewer #3: This manuscript reports on the effect of continuous and pulsed drought on amino acid, sugar and organic acids composition of different organs of wheat plants at two time points. Furthermore, they investigated the effect on aphids feeding on the respective plants and plant organs.

The manuscript is well written and the experiment is generally well explained and interpreted.

REPLY: Thank you for the positive feedback and constructive comments.

I only have a few comments and suggestions:

Under drought stress, plants tend to increase the number of trichomes to reduce water loss. Additionally, trichomes are an efficient morphological defense against small herbivores moving on the plant surface, like aphids. I wonder why the authors did not report any observations on trichome density.

REPLY: Unfortunately, we did not measure trichome density. However, we agree that possible differences in morphological defences caused by different irrigation regimes might play an important role for aphid development and added that aspect to our discussion (line 592).

You analyzed the concentrations of organic acids but the manuscript contains no explanation why they are relevant for the aphids.

REPLY: In the originally submitted manuscript we entitled the organic acid “aconitic acid” as amino acid in the discussion (lines 500 & 502). We apologise for the mistake and changed the term to “organic acid”. In addition to the example of a negative impact of an organic acid in the Zea mays-Rhopalosiphum padi system (cited in lines 524-27x), we added now a further recent study by Knoll et al. (2021) that found a negative correlation between aphid damage and trans-aconitic acid concentration (lines 527-530).

Introduction:

l. 104: why do you expect less population growth on stressed plants compared to well-watered plants when you explained in lines 70-74 that amino acid and sucrose concentrations may be increased?

REPLY: Different studies found evidence for different outcomes (i.e., positive, neutral or negative effects of drought). Thus, probably all hypotheses can be justified. Nevertheless, we changed now our hypothesis in line with the expectations for the chemical changes, expecting a better population growth and survival on stressed plants (lines 115ff). 

Material & Methods:

How did you determine the irrigation treatments? Have there been any pre-experiments? Why did you choose a soil water content of 23 and 11%, respectively?

REPLY: Yes, there were several pre-experiments in which we determined the irrigation treatments under the given conditions in the greenhouse. The soil water content of 23% was chosen since wheat plants showed optimal growth in the pots. Under a soil water content of 11%, plants showed wilting in between the irrigation events but recovered after watering (now also mentioned in lines 172-175). 

L. 156-160: I am quite surprised to see that you published data from the same experiment in two different articles. Usually, that should have been reported together shouldn’t it?

REPLY: In the already published paper, we focused entirely on the plant responses and show data of the whole leaf metabolome and of the water use efficiency of wheat plants under the given conditions. The present study focuses on the phloem sap metabolome and consequences of drought events for potential pest organisms feeding on phloem sap. Therefore, we decided to publish these two datasets separately. 

Your irrigation treatments are quite complex and not very easy to understand. Maybe it would help to add an explanatory figure? 

REPLY: We thank you for the suggestion and added a new figure (Fig 1) with an overview of the timeline of the experiments (irrigation scheduling, harvesting time points, start and end of aphid bioassays). Moreover, we revised the text of the method section for clarification.

Also, I am wondering for how many days the treatments continued: l. 152: “after this period…”: for how long?

REPLY: The irrigation treatments continued until the last aphid of the survival assays had died (day 155 after sowing; the T2 survival assay started 94 days after sowing and lasted for 61 days until the last aphid was dead). We added that information to the method part (line 299) and implemented it into the new Figure 1.

Results:

Figure 1 is quite difficult to read because of the large fraction of overlay. I do appreciate the value in presenting all the data together, as an overview. However, in the current version you cannot see much more than that the samples are all quite similar. I suggest to create three figures each for amino acids and for organic acids, in a similar way as figure 3: one for T1-leaves, one for T2-leaves, and one for T2-ears.

REPLY: We thank reviewer 3 for the advice. In the new Figure 2 we decided to show an overview for all treatments for amino acids (A) and organic acids with sugars (B), respectively. Additionally we show the suggested NMDS plots for T1 leaves (C&D), T2 leaves (E&F) and T2 ears (G&H) separately. We also modified the corresponding results part and address differences in the metabolite profiles between the treatment groups within time points and plant parts in more detail now by referring to the specific NMDS plots (lines 352ff).

Also, I don’t think grey was the best choice of color for the control because in some parts, it is hard to distinguish it from the dark purple.

REPLY: In the new Figure 2, convex hulls of the ctr groups are hatched to stand out against the two drought stress groups.

Figure 3: please indicate significant differences between the treatments. I suggest naming the y-axis “number of aphids”.

REPLY: The generalised linear model we used indicates whether the irrigation treatment, the plant part and/or the interaction of these terms significantly influence the population sizes of aphids. These results are given in the respective table. We now also tested for significant differences between certain groups (ctr vs. cd, cd vs. pd) using the contrast function in R (information added in method section, lines 337-340and 346) and implemented the results in the new Figure 4 and the results text (lines 423-424). We changed the label of the y-axis to “number of aphids” as suggested.

Discussion:

Your time points contain two factors that differ. While you do mention this at one point or the other in the manuscript, I think it might be helpful to summarize it (additionally?) in one place. Between T1 and T2, the duration of the drought treatments progresses (short/long term) alongside plant phenology (vegetative/reproductive stage). Hence, both could explain observed changes.

REPLY: We now additionally address this topic in the final conclusion (lines 613-615).

l.431: please add that the accumulation is problematic from a human perspective

REPLY: We deleted that aspect from the discussion, because Reviewer 1 noted that it might not be relevant to the study.

l. 496-498: Yes. Furthermore, this may also be important for the aphid population dynamics. Considering that you only observed aphid development for 13 days and that the pulsed drought treatment was watered every 8 days, it might be very relevant to show on which of these 13 days watering events occurred. It may have influenced your results greatly!

REPLY: Yes, this is an important point. In the new version of the manuscript we added a new Figure 1 in which the watering events and the timing of the bioassays are highlighted.

l.511: “specialized metabolites”: that is quite vage. Do you mean secondary metabolites? Plant defense metabolites?

REPLY: “Secondary” metabolites are nowadays usually called “specialised” metabolites. For clarification, we added “secondary” in parentheses (line 582).

l.514: here you should introduce the abbreviation BXD

REPLY: Since benzoxazinoids are only mentioned twice in the whole manuscript, we do not use the abbreviation BXD in the revised version anymore.

l. 518: do you have a reference for the detrimental effect of BXDs on aphids?

REPLY: We added the references Givovich et al. 1994 and Maag et al. 2015. 

l. 521: what are milk stages? Please elucidate

REPLY: we deleted the term “milk stages” and now precisely describe the stages (lines 596).

---

## [Editor Report · Decision Letter 1]

4 Jan 2022

Time point- and plant part-specific changes in phloem exudate metabolites of leaves and ears of wheat in response to drought and effects on aphids

PONE-D-21-29624R1

Dear Dr. Müller,

We’re pleased to inform you that your manuscript has been judged scientifically suitable for publication and will be formally accepted for publication once it meets all outstanding technical requirements.

Kind regards,

Nicolas Desneux

Academic Editor

PLOS ONE
---

## [Editor Report · Acceptance letter]

17 Jan 2022

PONE-D-21-29624R1 

Time point- and plant part-specific changes in phloem exudate metabolites of leaves and ears of wheat in response to drought and effects on aphids 

Dear Dr. Müller:

I'm pleased to inform you that your manuscript has been deemed suitable for publication in PLOS ONE. Congratulations! Your manuscript is now with our production department. 

Kind regards, 

on behalf of

Dr. Nicolas Desneux 

Academic Editor

PLOS ONE